# A Novel Classification Model for Lower-Grade Glioma Patients Based on Pyroptosis-Related Genes

**DOI:** 10.3390/brainsci12060700

**Published:** 2022-05-28

**Authors:** Yusheng Shen, Hao Chi, Ke Xu, Yandong Li, Xisheng Yin, Shi Chen, Qian Yang, Miao He, Guohua Zhu, Xiaosong Li

**Affiliations:** 1Department of Neurosurgery, The First Affiliated Hospital of Xinjiang Medical University, Urumqi 830054, China; yushengalex@outlook.com (Y.S.); drliyandong@163.com (Y.L.); 2Clinical Medicine College, Southwest Medical University, Luzhou 646000, China; chihao7511@163.com (H.C.); yxs15181529958@163.com (X.Y.); 3Department of Oncology, Chongqing General Hospital, Chongqing 401147, China; nsmcxuke@163.com; 4Clinical Molecular Medicine Testing Center, The First Affiliated Hospital of Chongqing Medical University, Chongqing 400016, China; krystalchen1999@163.com (S.C.); qian_yang125@163.com (Q.Y.); 5Laboratory Animal Center of Chongqing Medical University, Chongqing 400016, China; hemiao9906@163.com

**Keywords:** lower-grade glioma, pyroptosis, immunotherapy, targeted therapy

## Abstract

Recent studies demonstrated that pyroptosis plays a crucial role in shaping the tumor-immune microenvironment. However, the influence of pyroptosis on lower-grade glioma regarding immunotherapy and targeted therapy is still unknown. This study analyzed the variations of 33 pyroptosis-related genes in lower-grade glioma and normal tissues. Our study found considerable genetic and expression alterations in heterogeneity among lower-grade gliomas and normal brain tissues. There are two pyroptosis phenotypes in lower-grade glioma, and they exhibited differences in cell infiltration characteristics and clinical characters. Then, a PyroScore model using the lasso-cox method was constructed to measure the level of pyroptosis in each patient. PyroScore can refine the lower-grade glioma patients with a stratified prognosis and a distinct tumor immune microenvironment. Pyscore may also be an effective factor in predicting potential therapeutic benefits. In silico analysis showed that patients with a lower PyroScore are expected to be more sensitive to targeted therapy and immunotherapy. These findings may enhance our understanding of pyroptosis in lower-grade glioma and might help optimize risk stratification for the survival and personalized management of lower-grade glioma patients.

## 1. Introduction

Glioma is the most prevalent primary malignant intracranial tumor and can be divided into glioblastoma (WHO grade IV) and lower-grade glioma (WHO grade II/III) based on the WHO grading system [1]. Compared with glioblastoma (GBM), patients with lower-grade glioma (LGG) have a relatively favorable prognosis. Nevertheless, LGG is still a heterogeneous subgroup with a wide range of survival times (From 1 to 15 years). Thus, a more precise stratification system is needed. Despite the enormous progress in the molecular pathology of glioma, treatment options are limited. Temozolomide (TMZ) is the only FDA-approved chemotherapeutic for newly diagnosed GBM [2]. In recent decades, Stupp protocols (maximal safe surgical resection, radiation, and temozolomide chemotherapy) remained the gold standard for GBM treatments [3]. However, TMZ can induce hypermutation in LGG associated with malignant progression to GBM. Therefore, it is better to find new therapeutic strategies for LGG.

Pyroptosis is a relatively new form of programmed cell death associated with various therapies, including radiation [4], chemotherapy [5], targeted therapy [6], and immunotherapy [7]. Increasing studies have demonstrated that pyroptosis plays an essential role in tumor development, but its impact on specific cancer types remains a mystery. For example, the acute inflammation induced by pyroptosis can play an anti-tumor effect by enhancing immune response, while chronic tumor necrosis under the central hypoxia region triggered by pyroptosis promotes tumor progression [8]. More specifically, the expression of GSDMD (an executor of pyroptosis) is significantly reduced in gastric cancer and promotes tumor proliferation [9], while in non-small cell lung cancer, GSDMD is highly expressed with a poorer prognosis. However, this correlation was not found in squamous cell lung cancer patients [10]. These findings indicate pyroptosis’ complex role in tumor development.

Notably, pyroptosis is a multistep process involving different pathways (canonical, non-canonical inflammasome, and alternative pathways). The overlap and potential cross-talk of the different pathways, characterizing the overall effects of pyroptosis-related genes (PRGs) rather than a single PRG, may be a more effective strategy for the understanding role of pyroptosis. Thus, we first systematically analyzed the genomic and transcript PRG alterations in this study and explored their potential cross-talk in LGG. Second, based on PRGs, we developed and validated a risk-stratification signature to assess the prognosis and drug sensitivity in LGG. This work may help to optimize clinical decision-making in targeted therapy and immunotherapy for patients with LGG.

## 2. Materials and Methods

### 2.1. Datasets

RNA-seq data and corresponding clinical information of LGG patients were downloaded from the TCGA database (https://portal.gdc.cancer.gov/) (accessed on 19 July 2021). Transcriptome profiles in normal brain tissues were obtained from the TCGA and GTEx project (https://gtexportal.org/home/) (accessed on 19 July 2021). We used gene transcripts per million (TPM) data for the subsequent data analysis. The “Combat” method is an empirical Bayes method, which can estimate parameters for location and scale adjustment of each batch for each gene independently [11]. Numerous previous studies have proven its effectiveness [12,13,14]. Therefore, we used the “Combat” method to remove the batch effect for batch effects between TCG-LGG and GTEx datasets. In addition, we downloaded copy number variation (CNV) data and somatic mutation data from TCGA and UCSC Xena Browser (http://xena.ucsc.edu/) (accessed on 19 July 2021). The normalized single-cell dataset (TPM values) was downloaded from the GEO database (GSE163108, GSE182109), GSE163108 was derived from CD3+ single-cell sequencing data from 31 gliomas (15 IDH-G vs. 16 GBM), GSE182109 was derived from single-cell sequencing data from 18 gliomas (16 GBM, 2 LGG). The immunotherapy cohorts were obtained from the IMvigor210 cohort (urothelial carcinoma) and the GSE78220 cohort (melanoma).

### 2.2. Identification of Differentially Expressed PRGs

Thirty-three pyroptosis-related genes were extracted from prior studies [15,16,17]. Considering the small number of normal brain samples, we also obtained transcriptome data of normal brain samples from the GTEx database. Expression data from the two datasets were then merged and normalized to fragment per kilobase million (FRKM) values. The “Limma” R package was used to identify differentially expressed pyroptosis-related genes (DEGs) with FDR < 0.05 and |log2FC| ≥ 1.

### 2.3. CNV and Somatic Mutation Analysis of PRGs

The R package “Rcircos” was used to plot the genomic location of CNVs. The waterfall plot of a mutational landscape was generated using the “maftools” package.

### 2.4. Consensus Clustering Analysis of PRGs

Consensus clustering methods determined the matrix to classify the TCGA-LGG samples (R packages “limma” and “ConsensusClusterPlus”). We evaluated the relationships between clinical features and clusters by the chi-square test and R package “survival.”. Principal component analysis (PCA) was conducted by the “prcomp” function in the “stats” R package. The gene expression patterns among clusters were visualized by R packages “pheatmap”.

### 2.5. Development and Validation of a PyroScore Prognostic Signature

We used Cox regression analysis to evaluate each gene’s prognostic value, and genes significantly associated with overall survival (OS) time were enrolled in further analysis. We then performed the least absolute shrinkage and selection operator (LASSO) Cox regression analysis to avoid overfitting to train the predictive model. Eight genes were finally identified, their coefficients were determined by multivariate Cox regression, and the minimum criteria decided the penalty parameter. The risk score formula was as follows:(1)$${\mathrm{Pyroscore}=\sum }_{i}^{8}\mathrm{Coefi}\;\times \;\mathrm{Expi}\;(\mathrm{Coef}:\;\mathrm{coefficient},\;\mathrm{Exp}:\;\mathrm{expression}\;\mathrm{level}\;\mathrm{of}\;\mathrm{the}\;\mathrm{gene})$$
where Pyroscore denotes the risk score based on pyroptosis-related genes, Coef represents each gene’s multivariate cox regression coefficient, and Exp represents the expression level of pyroptosis-related genes.

TCGA-LGG samples were randomly divided into training cohorts and validation cohorts. Patients were grouped according to the median risk score, and their survival time was compared using Kaplan–Meier analysis and log-rank test. We also conducted time-dependent ROC analysis by the “survivalROC” R package. A forest plot was used to display the significance of each variable (risk score and clinical information) on prognosis.

### 2.6. Functional Enrichment Analysis and Immune Inflation Cell Analysis

DEGs among the high-risk and low-risk groups were identified based on the filter criteria (|log2FC| ≥ 1 and FDR < 0.05). KEGG and GO analysis of the DEGs were performed by using the “clusterProfiler” package, and the score of immune infiltrating cells and the activity of immune-related pathways were calculated using the ssGSEA method (“GSVA” R package).

### 2.7. TME Cell Infiltration Estimates

We performed ssGSEA to quantify the immune cell infiltration levels based on gene expression profiles. The specific marker genes of each immune cell were obtained from a previous study [18]. We calculated the stromal, immune, and ESTIMATE scores of all samples using the ESTIMATE algorithm [19].

### 2.8. Prediction of Therapeutic Response with High- and Low-PyroScore

We used the IMvigor210 cohort [20] and the GSE78220 cohort to evaluate different outcomes with immunotherapy. Based on the Genomics of Drug Sensitivity in Cancer (GDSC) database (https://www.cancerrxgene.org) (accessed on 19 July 2021), the half-inhibitory concentration (IC50) value of chemotherapies or targeted drugs were calculated via “pRRophetic” R package [21,22]. The R (Version 3.6.3) (https://www.r-project.org) software was used to perform the analysis.

### 2.9. Single-Cell Analysis of PRGs

We further analyzed 15 IDH-G and 2 LGGs from the single-cell dataset (GSE163108, GSE182109). The expression of model genes was analyzed in different cell subtypes after PCA analysis, cell clustering, and UMAP downscaling using the Seurat v3 R package. First, 2000 differentially expressed genes were selected for subsequent analysis using FindVariableFeatures, and gene expression was normalized using ScaleData. Then, principal component analysis was performed by RunPCA, and the top 15 were selected for clustering and UMAP downscaling. The clusters were determined by the expression of classical immune cell markers combined with detailed information from the original literature [23,24], where PTPRC is an immune cell marker; CD3D is a T cell marker; CD8A is a CD8+ T cell marker; CD4 is a CD4+ T cell marker; CD4, IL2RA, and FOXP3 are Treg cell markers; and CDK1 and NUSAP1 are cell cycle markers. Each cell marker is as follows: myeloid cells (expressing PTPRC/CD45, ITGAM/CD11B, and CD68), glioma cells (expressing SOX2, OLIG1, GFAP, and S100B), T cells (expressing PTPRC/CD45, CD3E, CD4, and CD8A), B cells (C11; expressing CD79A and CD19).

## 3. Results

### 3.1. The Landscape of Genetic and Expression Alterations of Pyroptosis-Related Genes in LGG

As shown in Figure 1A, the genetic mutation was only found in 32 of the 506 samples (6.32%), and no genetic alterations reached more than 2%. The CNV location is presented in Figure 1B, and further CNV frequency analysis showed prevalent CNV alterations among PRGs (Figure 1C). We observed that LGG samples could easily be distinguished from normal brain tissues according to the expression level of 33 PRGs (Figure 1D). 

The mRNA expression level of PRGs between LGG and normal samples was then investigated to ascertain whether these genetic alterations affected the mRNA expression of the PRGs (Figure 1F and Appendix A). Our results indicated that PRGs with CNV gain demonstrated significantly higher expression in LGG tissues than in normal brain tissues (e.g., GSDMC and TIRAP) and vice versa (e.g., NLRP2 and SCAF11). However, we also noted that several PRGs presented inconsistent CNV and mRNA expression changes (e.g., GSDMD and NLRP3). We speculated that CNV was not the only factor to regulate gene expression. Other factors, such as histone modification and DNA methylation, can also regulate gene transcription [25,26,27]. Notably, gene expression differences in some of the genes could also be influenced by the molecular type of LGG. For example, genes located on chromosome 19q (oligodendroglioma, molecular type: IDH-mutated, 1p/1pq-codeleted) and genes located on 7p (NOD1, GSDME, IL6) could be altered by frequent 7p gains in molecular GBM. Besides, we also found potential cross-talks between 33 PRGs (Appendix A). 

### 3.2. Identification of Tumor Cluster Pattern by Consensus Clustering

To characterize the relationships between PRGs and LGG subtypes, we then performed a consensus clustering analysis based on gene expression patterns. The consensus matrix heatmap showed the preferable sharp boundaries at K= 2, indicating stable samples clustering (Figure 2A). Based on the Kaplan–Meier curves, Cluster 2 had a longer survival time than Cluster 1 (Figure 2C). We also found a difference in the clinical characteristics of the two clusters (Figure 2B).

### 3.3. Development and validation of a PyroScore

The TCGA-LGG samples were randomly split into training and validation cohorts. We first used univariate Cox regression analysis to screen the survival-related genes. A total of 20 genes were identified as significantly associated with survival time. (Appendix A, *p* < 0.05). 

We then fitted a regression model by employing the LASSO method, and an 8-gene signature was finally constructed (Figure 3A,B). The risk score formula was obtained as follows: risk score = (−0.0983) × TNF + (−0.116) × TIRAP + (−0.621) × CASP9 + (0.799) × PLCG1 + (0.193) × PRKACA + (0.127) × CASP3+ (0.0685) × CASP8+ (0.474) × CASP4. Patients were stratified into high-risk and low-risk groups according to the risk score (Figure 3C). An increasing risk score was associated with more death events (Figure 3D). Significant survival difference was observed between the two groups (Figure 3F). Further time-ROC analysis showed the sensitivity and specificity of the model with AUCs of 0.896 (1-year), 0.893 (3-year), and 0.832 (5-year) (Figure 3G). In the validation cohort, this model had AUCs of 0.879 (1-year), 0.871 (3-year), and 0.672 (5-year) (Appendix A). Kaplan–Meier analysis also demonstrated significantly worse survival in the high-risk group (Appendix A).

We also performed a subgroup survival analysis based on 2021 WHO classification [28]. The subgroup analysis demonstrated the predictive value of pyroscore in astrocytoma, oligodendroglioma, and IDH-wt patients (Figure 4).

### 3.4. Clinical Characters and Prognosis Value of PyroScore in LGG

We conducted a univariate and multivariate analysis to assess whether the pyroptosis-related prognostic signature was independent of other clinical prognostic factors. The univariate analysis showed that pyroptosis-related prognostic score was an essential indicator in training and validation cohorts (Figure 5A,C). Multivariate analysis revealed that the pyroptosis-related prognostic score was a solid independent prognostic factor (Figure 5B,D). As demonstrated in Figure 5E, there were considerable differences between the high-risk and low-risk groups regarding age, grade, histology, and survival status. Strikingly, significantly more IDH mutation events were found in the low-risk group than in the high-risk group. Given the importance of clinical features, we also established an integrative nomogram that combined the pyroptosis-related prognostic signature and clinical factors (Appendix A). The calibration curve indicated that the bias-corrected curve fitted the apparent curve in the whole cohort relatively well (Appendix A).

### 3.5. PyroScore Is Associated with Immune Infiltration in LGG

We conducted the single-sample gene set enrichment analysis (ssGSEA) to compare the enrichment levels of immune cells and pathways between the high-risk and low-risk groups. In the TCGA-train cohort (Figure 6A), a higher level of immune cell infiltration was observed in the high-risk group than in the low-risk group, especially pDCs, T-cell-co-inhibition, T-cell-co-stimulation, TILs, Tregs, and T-helper cells (Th2, Th3, Th6). Most of the immune pathways were upregulated in the high-risk group, including B cells, cytokine-cytokine receptor, APC co-inhibition, APC co-stimulation, checkpoint, type-1 IFN responses, and type-2 IFN responses. Similarly, these findings were also found in the TCGA-validation cohort, except for NK cells (Figure 6B). Our results revealed that Nk cells were significantly higher in the low-risk group than in the high-risk group in the TCGA-train cohort. We also found that the DEGs were significantly involved in tumor immunity (Appendix A).

Considering the known effect of IDH mutation on the tumor immune microenvironment [29,30], we next performed a subgroup ssGSEA analysis based on IDH status (Figure 6C,D). The subgroup analysis indicated that most differences in tumor immune microenvironment still existed in IDH-mut and IDH-wt groups except T helper cells (Th3, Th6). Correspondingly, we did not find specific distributions of model genes and Pyroscore in CD3+ single-cell sequencing data (Appendix A). However, we found that most PRGs were expressed explicitly on myeloid cells (Figure 7). We then put PRGs as a gene module to calculate the score using AddModuleScore methods, and the results also indicated that PRGs were expressed explicitly on myeloid cells (Figure 8A–D). The 33 PRGs models could be well represented by 8 PRGs models (Figure 8E). The above data demonstrated that PyroScore could stratify the immune phenotypes of LGG.

### 3.6. Characteristics of the TME in the Different PyroScore Risk Groups

Compared with the low-risk group, almost all the costimulus molecules were upregulated in the high-risk group (Figure 9A). Elevated Enrichment Scores (StromalScore, ImmuneScore, ESTIMATEScore) were found in the high-risk group, representing a higher content of immune and stromal cells in the TME (Figure 9C). We further analyzed the correlation between the risk score and immune microenvironment cells. The results demonstrated a positive correlation between the score and nearly all immune microenvironment cells, particularly central memory CD8 T cells and Type 1 T helper cells (Figure 9B). A significant association was also found between the immune cells and the eight model genes (Figure 9D).

### 3.7. Distinct Somatic Mutation Status in the High- and Low-Risk PyroScore Group

Our analyses suggested a specific somatic mutation distribution among two risk groups. As shown in Figure 10A, IDH1, TP53, ATRX, EGFR, TTN, PTEN, and NF1 were the most frequently mutated genes in the high-risk PyroScore group, while IDH1, TP53, ATRX, CIC, FUBP1, PIK3CA were the top mutated genes in the low-risk PyroScore group (Figure 10B). We also observed that the frequency of IDH1 mutations was significantly higher in the low-risk PyroScore group than in the high-risk PyroScore group (93% versus 40%), which is consistent with previous studies IDH-mutant mutation patients have a better prognosis [31,32].

We next analyzed the association between risk score and tumor mutation burden (TMB). Patients in the high-risk group harbored a higher tumor mutation burden (Figure 10C). Meanwhile, the PyroScore was positively corrected with TMB (Figure 10D).

### 3.8. PyroScore May Be an Effective Factor in Predicting Potential Therapeutic Benefits

Previous studies have confirmed TMB as a valuable response biomarker for immunotherapy in various cancers [33,34,35], which indicated that PyroScore might also be a potential predictive marker in patients with immunotherapy. Hence, we next explored whether PyroScore could predict the efficacy of immunotherapy. In the anti-PD-L1 cohort (IMvigor210: urothelial carcinoma patients), patients with a low PyroScore exhibited significantly prolonged survival (Figure 11A), and the objective response rate was higher in the low-risk score group (Figure 11B). We did not find a significant difference in the anti-PD-1 cohort (GSE78220: melanoma patients) (Figure 11C,D), likely due to the limited sample size.

Based on the GDSC database, we evaluated drug sensitivity between different risk score groups. The results showed that patients with low PyroScore were more sensitive (lower IC50) to ABT-263, ABT-888, AG-014699, AICAR, AMG-706, and ATRA (Figure 12). Therefore, we hypothesized that pyroscore might be an effective factor in predicting potential therapeutic benefits. Notably, the above in silico data warrants further validation in vitro, in vivo, and prospective trials.

## 4. Discussion

Both LGG and GBM belong to glioma, but they exhibit different clinical characteristics and biological behaviors. Clinically, LGGs are more frequently seen in highly functional areas of the brain [36], and seizures have become more prevalent [37]. LGG has a more indolent biological behavior than GBM. The recent theory of immune normalization has considered altering the tumor immune environment to be a gradual process, not an event. Based on the above summary, LGG might be more effective for immunotherapy than GBM. Indeed, there has been considerable evidence that LGG has a unique tumor immune microenvironment, which can be distinguished from GBM [38]. Multiple immunotherapy clinical trials have failed in GBM, but there is a relative paucity of clinical trials in LGG precisely because of its longer survival time, which leads to higher research and development costs.

Pyroptosis has recently emerged as an exciting area. To date, some pyroptosis-related gene signatures have been developed in multiple tumor types, including bladder [39], ovarian [40], and colorectal cancer [41]. Not unexpectedly, several studies have constructed pyroptosis-related gene signatures in glioma [42,43], but all took lower-grade glioma and glioblastoma as a single entity. Crucially, the impacts of pyroptosis on the tumor immune microenvironment and therapeutic response have been poorly studied in the above studies.

In this study, we first systematically analyzed the alterations of 33 PRGs at the transcriptome and genomic levels in LGG. The results showed an imbalanced expression of PRGs between LGG and normal tissues. We identified two distinct pyroptosis clusters with significantly different survival and clinical characteristics. Based on the univariate Cox regression analysis and LASSO regression analysis, we constructed an 8-gene signature (TNF, TIRAP, CASP9, PLCG1, PRKACA, CASP3, CASP8, CASP4) called PyroScore. 

Among these signatures (TNF, TIRAP, CASP9, PLCG1, PRKACA, CASP3, CASP8, CASP4), TNF (Tumor necrosis factor) is currently considered a two-edged sword in cancer development. On the one hand, TNF can promote cancer as an endogenous tumor promoter. On the other hand, TNF is capable of activating cell pyroptosis by caspase-8 [44]. Our results showed that TNF was highly expressed in LGG, but positively, with a better prognosis. TIRAP is a signaling adaptor associated with Toll-like Receptor (TLR)-mediated innate immune signaling. While TIRAP has long been thought to affect antimicrobial immunity, recent work indicated that TIRAP can also be involved in tumorigenesis [45]. In the present study, we found that a high expression of TIRAP was associated with a more favorable prognosis. Variants of CASP9 have shown a strong association with glioma. A germline stop–gain mutation (R65X) was identified in a family with Li-Fraumeni-like syndrome. This mutation generated a short CASP9 isoform and may disrupt the p53 signaling pathway [46]. Additionally, whole-exome sequencing revealed another stop-gain mutation in pediatric astrocytoma [47]. Our study indicated that CASP9 overexpression was beneficial to prognosis. The Phospholipase C gamma 1 (PLCG1) gene is a member of the PLC superfamily. A previous study found that GSDMD N-terminal-mediated pyroptosis is dependent on the PLCG1 gene in sepsis [48]. Consistent with our results, a high expression of PLCG1 predicts poor survival in adult lower-grade gliomas. PRKACA belongs to the PKA signaling pathway; an elevated expression of PRKACA can regulate HER2-targeted therapy in breast cancer cells through the inactivation of the pro-apoptotic protein BAD [49]. CASP8 is an extremely intriguing molecular switch for apoptosis, necroptosis, and pyroptosis [50]. The present study revealed that CASP8 expression was strongly positively associated with the activation of most immune cells, suggesting the contributory role of CASP8 in regulating tumor immunology in LGG. CASP3 is a classical executioner caspase, and recent investigations uncovered another inflammatory effect of CASP3 by cutting GSDME [51]. In our study, CASP3 was highly expressed in LGG and was an unfavorable prognostic factor. Distinct from CASP3, CASP4 induces pyroptosis by cleaving GSDMD [52]. A previous study demonstrated that CASP4 could activate CASP1 in inflammation [53]. Our results showed a close relationship between CASP4 and CASP1 at the transcriptome and genomic levels in LGG. The overexpression of CASP4 indicated a worse prognosis.

Unlike apoptosis, Pyroptosis is characterized by cell membrane rupture, followed by amounts of proinflammatory cytokines being released [8]. Recent studies indicated that the pyroptosis-related gene (GSDME) could suppress tumor growth by activating anti-tumor immunity [54]. However, the relationship between pyroptosis and the tumor immune microenvironment in LGG is still not well-understood. In our study, PyroScore was positively associated with most immune cells. Patients with a high PyroScore showed higher immune cell infiltration but worse prognosis, in contrast with previous studies on other cancer types [55,56]. A possible explanation for this is that high-PyroScore patients were characterized by both immune-hot and immune-suppressive phenotypes. We observed an elevated expression of immune-suppressive cells in the high-PyroScore group (e.g., Treg, macrophage). The above results are interesting but not surprising. In accordance with the present results, previous studies demonstrated that immune-hot gliomas harbored an abundance of immune-suppressive cells with a poorer prognosis [57,58]. We further explored the expression of PRGs in different cell types. While T cells play a significant role in generating anti-tumor effects, we did not find specific expression of PRGs in T cells but found PRGs were expressed explicitly in myeloid cells. In fact, the myeloid cell is the largest part of the immune cell compartment in glioma [59], which plays an essential and complex role in the tumor immune microenvironment [60]. As previously stated, pyroptosis is a double-edged sword in the tumor immune microenvironment. Our study demonstrated that pyroptosis in myeloid cells shapes an inflammatory and immune-suppressive microenvironment, thus promoting the development of lower-grade glioma, which is consistent with previous studies in pancreatic carcinoma [61] and breast cancer [62]. All these findings indicated that our risk model could assess the tumor immune microenvironment in LGG.

Consistently, the high-risk group presented with an immune-suppressive tumor microenvironment. Given this, we further examined the efficacy of immunotherapy among the two risk groups in both the anti-PD-L1 cohort and anti-PD-1 cohort. Unexpectedly, the low-PyroScore group showed a better response to anti-PD-L1 than the high-PyroScore group but had a lower mutational burden. These findings are consistent with earlier studies, indicating that a low tumor mutation burden is associated with better survival when treated with recombinant polio virotherapy or an immune checkpoint blockade in recurrent glioblastoma patients [63]. Due to the relatively small sample size, we did not find survival differences in the anti-PD-1 cohort. 

At present, pharmacotherapeutic options for LGG are still limited. This model identified six potential drugs for low-PyroScore patients, including ABT-263, ABT-888, AG-014699, AICAR, AMG-706, and ATRA. ABT-263, also known as Navitoclax, is a small-molecule Bcl-2 inhibitor, which could effectively induce apoptosis [64]. The previous studies found that IDH1-mutated gliomas are particularly vulnerable to ABT-263 [65]. Similarly, our study found that patients with low PyroScore showed more percentage of mutant IDH gene. AG-014699 (rucaparib) is a PAPR inhibitor approved to treat recurrent ovarian cancer, fallopian tube carcinoma, and primary peritoneal cancer [66]. Notably, several studies have suggested that rucaparib has a limited ability to pass the blood–brain barrier [67]. ABT-888 (Veliparib) is another PARP inhibitor that can cross the blood–brain barrier [68]. A multicenter randomized phase II clinical trial (the VERTU study) found that the veliparib-containing regimen dose did not prolong survival time in glioblastoma, although it is tolerable; further correlative analysis should be performed to identify subpopulation benefits from veliparib [69]. The present study indicated that low-PyroScore patients were more sensitive to PAPR inhibitors (AG-014699, ABT-888). AICAR is a direct AMPK agonist, showing therapeutic potential for glioma, independent of AMPK [70]. Our results showed that the AICAR’s therapy response was associated with pyroptosis. Further investigation is needed to clarify the relationship between AICRA and pyroptosis. AMG-706 (Motesanib) is an oral multi-kinase inhibitor that selectively targets VEGFR, KIT, RET, and PDGFR [71]. The phase III Motesanib trial did not improve progression-free survival (PFS) in non-small cell lung cancer patients. However, a recent study demonstrated that the combined treatment of Motesanib and Panitumumab is a promising strategy for glioma [72]. All-trans retinoic acid (ATRA) is known for the treatment of acute promyelocytic leukemia. Interestingly, a previous study found that ATRA specifically enhanced terminal granulocytic differentiation and shrunk tumor burden in IDH-mut AML cell line and xenografted mice model [73]. In our research, the low-risk PyroScore group carried a significantly higher mutant frequency of IDH, exhibiting a higher sensitivity to ATRA. In summary, the above findings suggested that our risk model could identify and select the patients who will respond to immunotherapy and targeted therapy. 

The PyroScore can be easily quantified by qPCR assay, suggesting a promising clinical translational value. Some limitations should be noted. First, this model was only constructed for in silico analysis, and some of the recognized prognostic factors were missed (such as the KPS score, the extent of surgical resection). Thus, a well-designed prospective multicenter study containing complete data should be performed to validate this scoring system in the future. Second, we could not classify the IDH-wt glioma further due to the missing molecular information (such as the TERT promoter mutation and EGFR amplification). Third, pyroptosis is a relatively new field of oncology. The roles of most PRGs in LGG remain unknown. Future wet-lab experimental studies addressing the complex functions are urgently needed. Fourth, there is currently no available LGG immunotherapy data for direct validating the predictive value of pyroscore. The tumor immune microenvironment in the IMvigor210 cohort (urothelial carcinoma) and GSE78220 cohort (melanoma) might differ from LGG. Further validation of the LGG immunotherapy cohort is needed.

## 5. Conclusions

In summary, our study compressively explored the genomic and transcript alterations of PRGs in LGG. The pyroptosis-based signatures could refine the LGG with stratified prognosis, distinct tumor immune microenvironment, and Pyscore may be a promising factor in predicting potential therapeutic benefits. Further validation in vitro, in vivo, and prospective trials are needed.

## Figures and Tables

**Figure 1 brainsci-12-00700-f001:**
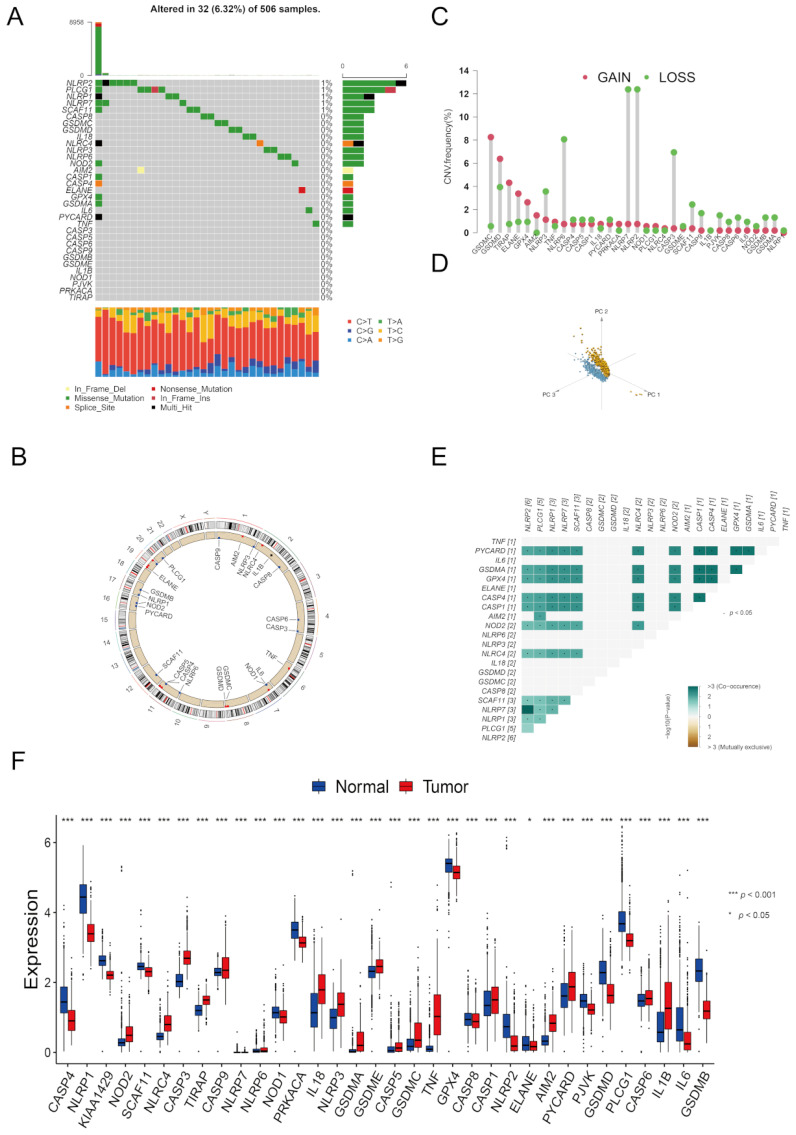
The genomic and transcript alterations of PRGs in LGG. (**A**) The mutation frequency of PRGs in LGG. (**B**) Location of CNV alteration of PRGs on 23 chromosomes in LGG. (**C**) CNV alteration frequency of PRGs in LGG. The deletion (amplification) frequency was marked with a green (red) dot. (**D**) PCA clusters. (**E**) Cross-talk between PRGs at the genomic level. (**F**) The expression difference of PRGs among LGGs and normal brain tissues.

**Figure 2 brainsci-12-00700-f002:**
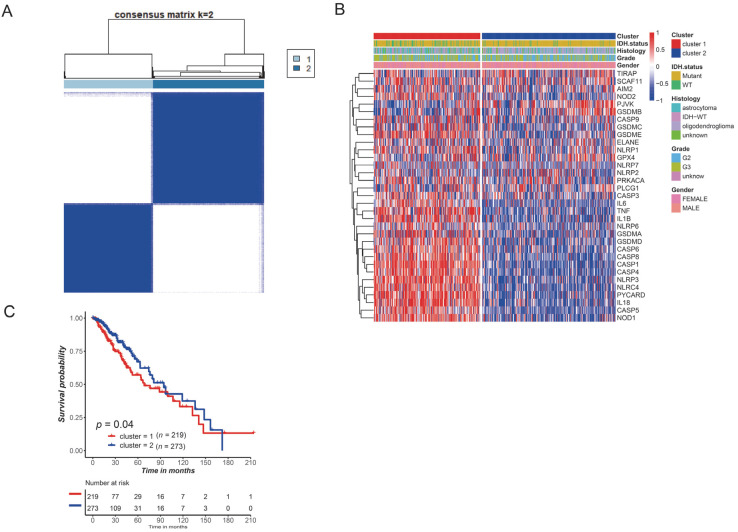
Identification of tumor cluster pattern. (**A**) The TCGA-LGG patients were divided into 2 cluster patterns (K = 2). (**B**) Different clinical characteristics among the 2 clusters. (**C**) Survival difference between 2 clusters.

**Figure 3 brainsci-12-00700-f003:**
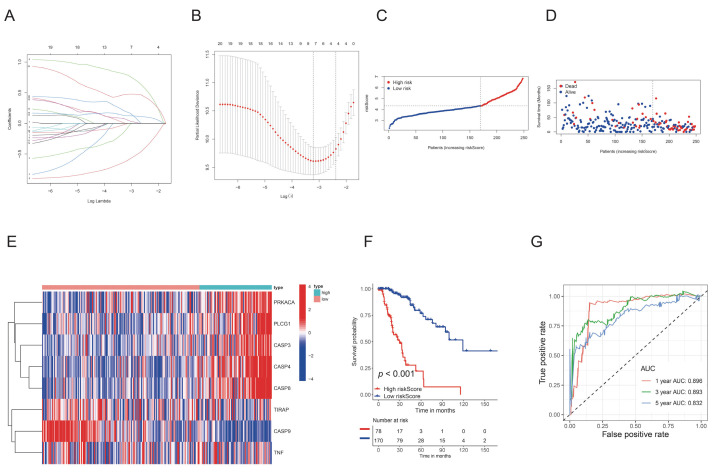
Development of the PyroScore. (**A**) The LASSO-Cox regression model was used to identify the most robust genes. (**B**) Cross-validation of parameter selected by LASSO. (**C**) Distribution of patients according to the PyroScore. (**D**) Survival status distribution of different risk patients. (**E**) Heatmap showing expression levels of 8 PRGs among two risk groups. (**F**) Overall survival difference among the two risk groups. (**G**) ROC curves measure the predictive value of PyroScore.

**Figure 4 brainsci-12-00700-f004:**
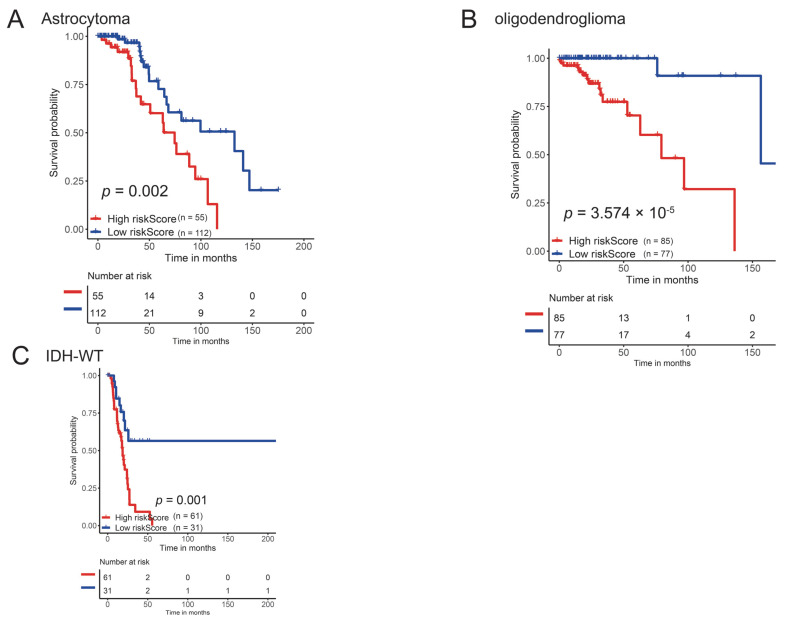
Subgroup survival analysis based on histology types. (**A**) Overall survival difference among two risk groups in astrocytoma patients. (**B**) Overall survival difference among two risk groups in oligodendroglioma patients. (**C**) Overall survival difference among two risk groups in IDH-wt patients.

**Figure 5 brainsci-12-00700-f005:**
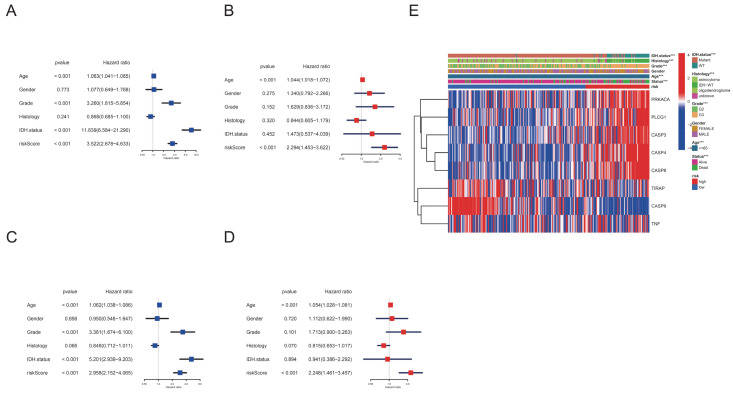
Univariate and multivariate Cox regression analyses for PyroScore. (**A**) Univariate analysis for the TCGA training cohort. (**B**) Multivariate analysis for the TCGA training cohort. (**C**) Univariate analysis for the TCGA validation cohort. (**D**) Multivariate analysis for the TCGA validation cohort. (**E**) Heatmap showing the association between clinical characteristics and the risk groups (*** *p* < 0.001).

**Figure 6 brainsci-12-00700-f006:**
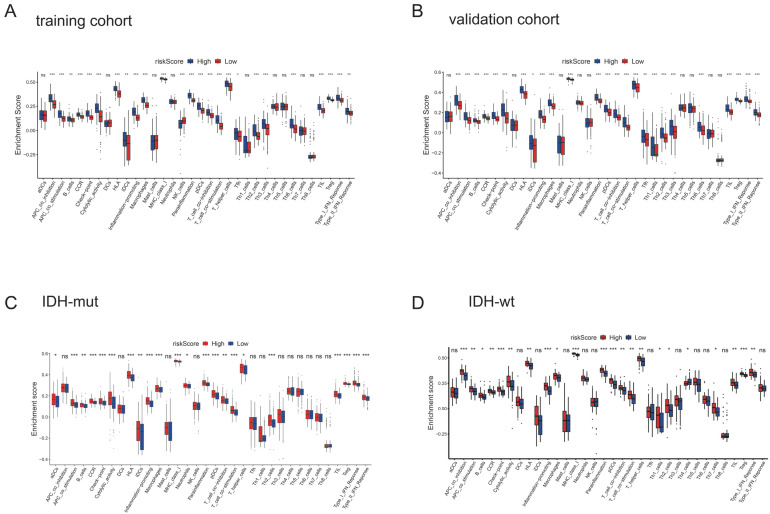
ssGSEA analysis of immune cells and immune pathways. (**A**) Comparison of the enrichment scores of immune cells and immune pathways between different risk groups in the training cohort. (**B**) Comparison of the immune cells and immune pathways in the validation cohort. (**C**) Comparison of the enrichment scores of immune cells and immune pathways between different risk groups in the IDH-mut cohort. (**D**) Comparison of the immune cells and immune pathways in the IDH-wt cohort. *p* values were shown as: ns not significant; * *p* < 0.05; ** *p* < 0.01; *** *p* < 0.001.

**Figure 7 brainsci-12-00700-f007:**
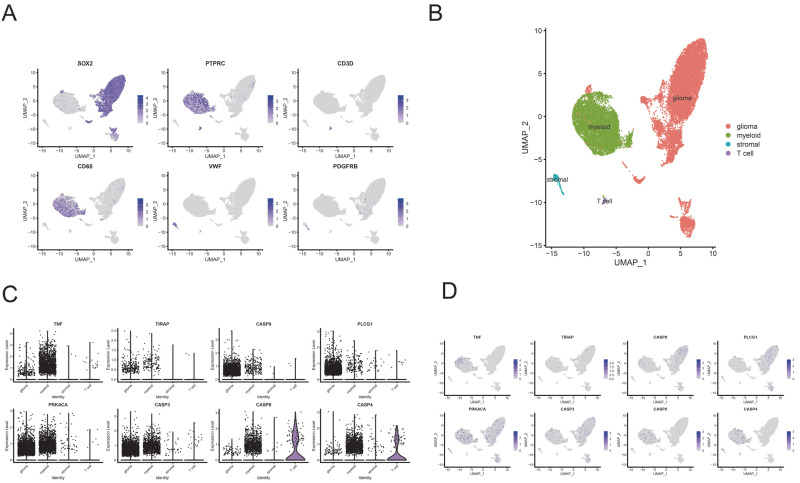
Single-cell analysis of PRGs (GSE182109). (**A**) UMAP plots for marker genes. (**B**) UMAP plots for all cells. (**C**) Violin plots of the expression of 8 target genes in 4 cell subtypes. (**D**) UMAP plots of the eight target genes’ expression.

**Figure 8 brainsci-12-00700-f008:**
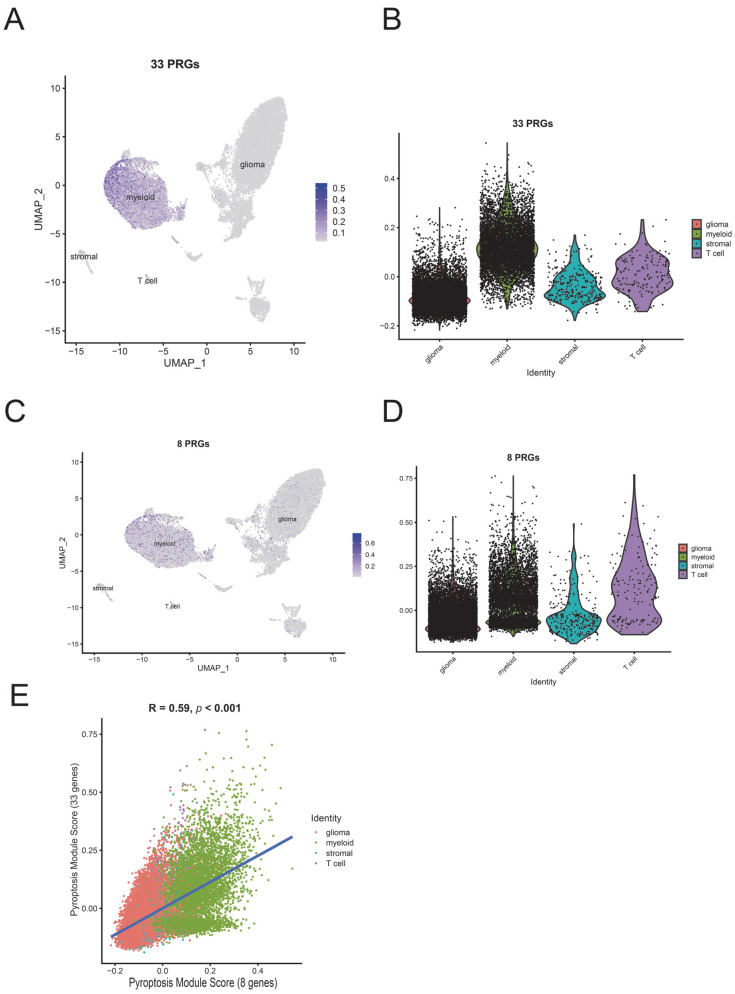
Single-cell analysis of PyroScore and model genes (GSE163108). (**A**) UMAP plot for 33 PRGs: The 33 target genes were used as a gene module to calculate the score using AddModuleScore. (**B**) Violin plots of the expression of the 33 gene module. (**C**) UMAP plot for eight PRGs: The eight target genes were used as a gene module to calculate the score using AddModuleScore. (**D**) Violin plots of the expression of the eight gene module. (**E**) Correlation between 33 PRGs modules and eight PRGs modules.

**Figure 9 brainsci-12-00700-f009:**
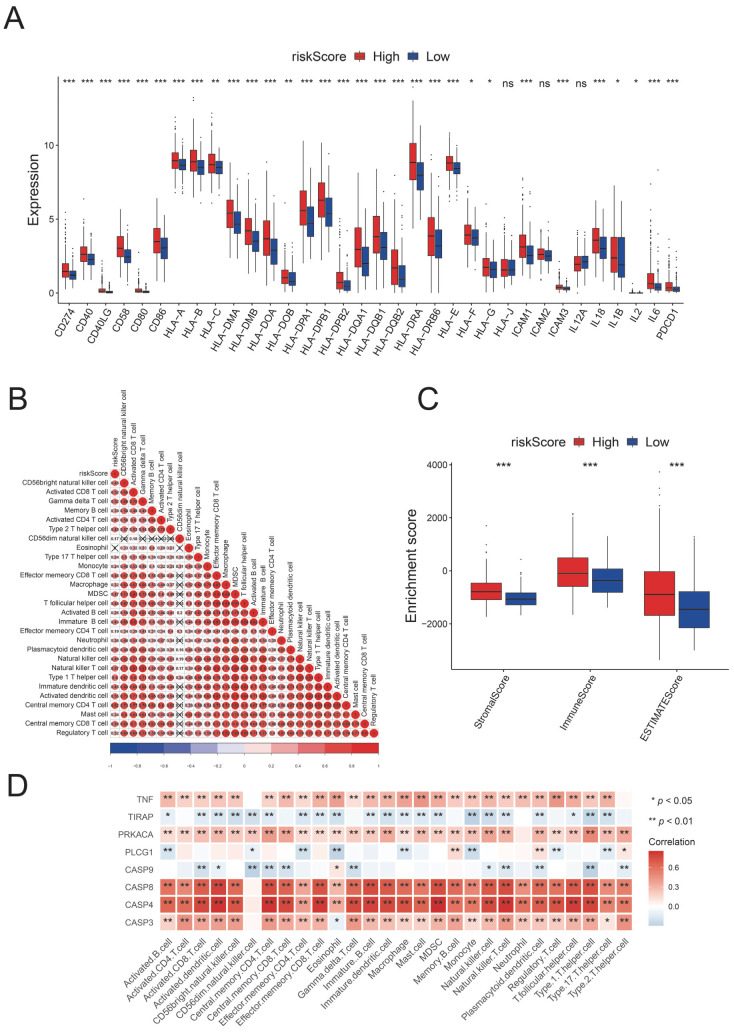
Characteristics of the TME in the different PyroScore risk groups. (**A**) Differences in MHC class I gene and immune checkpoint gene expression between different risk groups. (**B**) Correlation analysis of PyroScore and immune cells. (**C**) Comparison of ImmuneScore, StromalScore, and ESTIMATEscore between the two risk groups. (**D**) The relationship between immune cells and eight model genes (ns not significant; * *p* < 0.05; ** *p* < 0.01; *** *p* < 0.001).

**Figure 10 brainsci-12-00700-f010:**
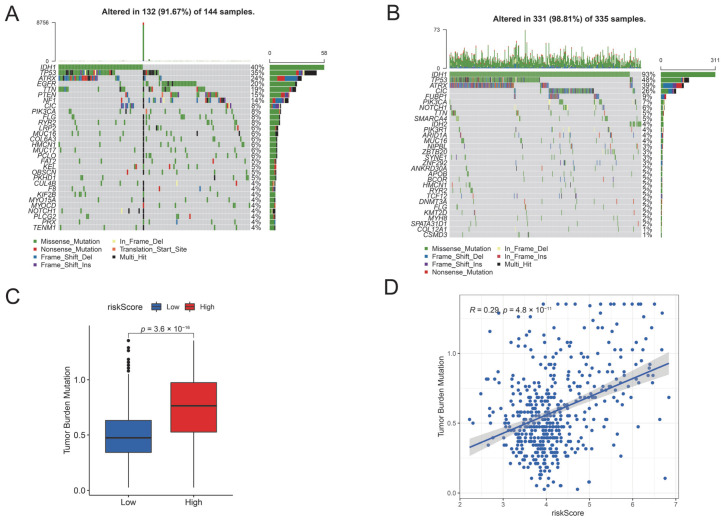
Distinct somatic mutation status in the different PyroScore groups. (**A**) Most frequently mutated genes in the high-risk group. (**B**) Most frequently mutated genes in the low-risk group. (**C**) Comparison of tumor mutation burden between the low-and high-risk group. (**D**) The correlation between PyroScore and tumor mutation burden.

**Figure 11 brainsci-12-00700-f011:**
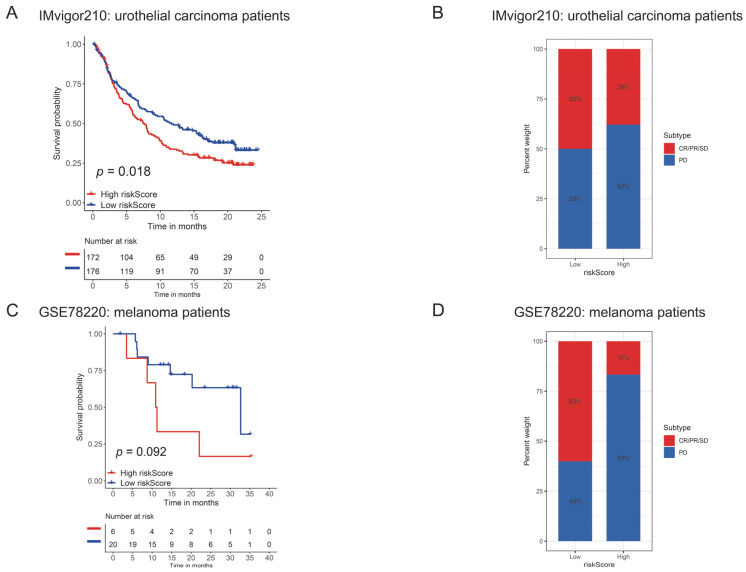
The efficacy of PyroScore in predicting the therapeutic benefits of immune-checkpoint blockade immunotherapy. (**A**) Kaplan–Meier survival plot showed a significant survival benefit in the low PyroScore group of the IMvigor210 cohort (urothelial carcinoma). (**B**) The proportions of clinical response in the low-and high-risk group of the IMvigor210 cohort (urothelial carcinoma). (**C**) Kaplan–Meier survival plot did not show a significant survival difference between the two risk-group of the GSE78220 cohort (melanoma). (**D**) The proportions of clinical response in the low-and high-risk group of the GSE78220 cohort (melanoma).

**Figure 12 brainsci-12-00700-f012:**
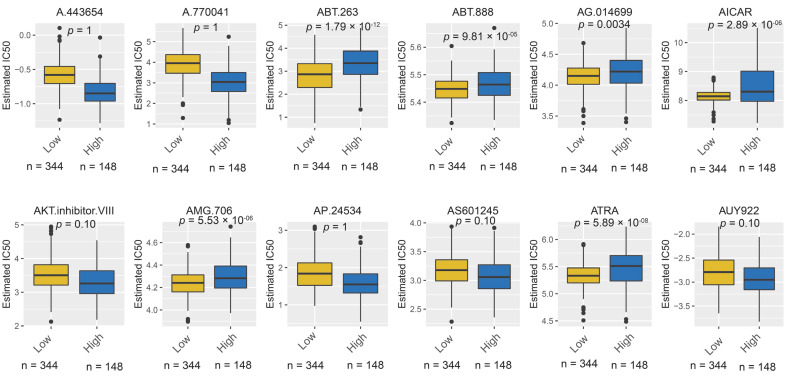
Estimated drug sensitivity in patients with high- and low-PyroScore group.

## Data Availability

The datasets analyzed in this study can be found in the TCGA-LGG project (http://www.cancer.gov/tcga) (accessed on 19 July 2021), GEO database (https://www.ncbi.nlm.nih.gov/geo/query/acc.cgi?acc=GSE163108), and GTEx project (https://gtexportal.org/home/).

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
