# Peer review of "A Novel Classification Model for Lower-Grade Glioma Patients Based on Pyroptosis-Related Genes"

_brainsci, 2022, doi:10.3390/brainsci12060700_

Round 1

Reviewer 1 Report

In this manuscript, Shen et al. evaluated differences in gene expression of pyroptosis-related genes between lower-grade glioma and normal brain tissue and identified two pyroptosis phenotypes in LGG. Based on individual differences and by applying a lasso-cox model, the authors developed a score displaying pyroptotic activity in patients. They found that the “PyroScore” correlates to survival, the immune microenvironment and therapy responses. The findings are interesting, but there are several conceptual issues that limit the quality, validity and impact of the presented work.

Major issues:

  1. Gliomas are a heterogeneous group whose biological behavior including genomic alterations and gene expression differs considerably between histological and molecular subtypes (IDH mutation, 1p19q codeletion etc.) according to the 2016/2021 WHO Classifications and this may also impact expression of pyroptosis-related genes. This fact was not appropriately addressed by the authors due to the following aspects:
    1. The authors did not correct any of their findings for molecular diagnoses according to the WHO classification.
    2. The authors included oligoastrocytomas in Figure 3B which are considered a deprecated entity since 2016.
    3. CNV variations and gene expression differences in some of the genes (NLRP2, NLRP7) are not surprising as they are located on chromosome 19q (which is by definition deleted in oligodendroglioma, IDH-mutated, 1p/1pq-codeleted). Similarly, gene expression of other genes located on 7p (NOD1, GSDME, IL6) may be altered by frequent 7p gains in molecular GBM.
    4. Also the immune microenvironment is known to differ between distinct molecular glioma subgroups with higher T cell infiltration in IDH wild-type as compared to IDH-mut glioma. Higher (derived from expression data) immune cell infiltration in the PyroScore high group may be related to IDH mutational status, as IDH mutations were found in 40% of PyroScore high group as compared to 93% of the PyroScore low group.
  2. Cox regression analysis for PyroScore: Important clinical factors with known impact on survival are missing (e.g. extent of resection, performance status). If these data are not available, the authors should at least acknowledge this as a limitation. 
  3. It is unclear why the authors used the IMvigor210 cohort (urothelial carcinoma) and the GSE78220 cohort (melanoma) to validate their score and estimate immunotherapy responses as their score was based on a glioma cohort. Indeed, these tumors differ considerably in terms of tumor-immune system interactions.
  4. Overall, the authors conclude that the PyroScore can “identify and select patients who will response to immunotherapy and targeted therapy”. The data do not support this conclusion. Without further validation of the involved genes on protein and/or cellular level and in prospective clinical trials, this in silico study remains hypothesis-generating and I do not see any clinical implications of their findings at the moment.

Minor issues:

  1. The included figures would benefit from some editing.
  2. The authors should improve language to improve readability.

Reviewer 2 Report

The manuscript is compelling and suggests that the chosen pyroptosis-related gene set could be the potential biomarker for the choice of glioma treatment.  A novel classification model for lower-grade glioma patients based on pyroptosis-related genes validated a risk stratification signature to assess the prognosis and drug sensitivity in low-grade gliomas. PyroScore can refine the lower grade glioma patients with a stratified prognosis, distinct tumor immune microenvironment, and different drug sensitivities. However, although the manuscript is interesting for a wide audience, the manuscript needs significant improvement; please see the attached recommendations.

1.         Line 153. Fig. 1F. The expression difference of PRGs among LGGs and normal brain tissues. Please provide details of the normalization of the expression levels of target genes across the LGGS and normal tissues. Data is not  normalized to the expression of home genes (like ACTB or RPS9). You have to provide the expression (or difference in the expression) of the home genes in the graph to be sure that their expressions are similar across LGGS and normal tissues. Please show the expression (or difference in the expression) of home gene/genes (as ACTB, RPS9) in the graph.

2.         Figure 2A and Figure 2C need better resolution; the gene names are not readable.

3.         Line 177. Based on the K-M curves. Please change K-M to Kaplan-Meier curves. Please indicate that the corresponding numbers of patients for each cluster are provided below the Kaplan-Meier graph.

4.         Fig.4A needs better resolution for the table and graph. Provide the table as a word file.

5.         Fig.4B,C, D,E need better resolution, axis titles are barely readable.

6.         Fig.6 is unreadable and needs better resolution.

7.         Fig.8 Indicate what parameter/score is presented on the horizontal axis.

8.         Supplemental figures should be presented in the supplemental file. Figure S1 should be moved to the supplemental file or renamed as the main figure.

9.         Fig. 13. Please indicate the number of patients (or cell lines?) per group for each compound. It looks like the data is presented based on cell lines’ sensitivity to the chemotherapeutic. Please clarify this issue.  

Round 2

Reviewer 1 Report

The authors responded to most of the raised concerns and significantly improved the manuscript. However, there are still some issues which should be considered to render the manuscript suitable for potential publication. 

Major issues:

  1. WHO classification/molecular phenotyping: The authors now appropriately distinguish between molecular glioma subtypes. Furthermore, it is a major asset of the revised version that a prognostic significance of the PyroScore could be observed in all molecular subtypes (Figure 6). However, I still suggest that oligoastrocytomas should be reclassified as either oligodendrogliomas or astrocytomas in Figure 3 and 7. As to the authors' question in Response 1b, I would like to point out that ATRX mutation is not a necessary condition for the classification of astrocytomas and the distinction between astrocytomas and oligodendrogliomas is exclusively done via 1p/19q codeletion and not by ATRX mutation (IDH mutation + 1p/19q codeletion = oligodendroglioma, IDH mutation + intact 1p/19q = astrocytoma irrespective of ATRX status).
  2. Single cell data analysis for immune microenvironment data according to molecular subtypes: This opens a new perspective to the work. Indeed, myeloid cells are major players in the immune microenvironment of gliomas as compared to other intracranial tumors such as brain metastases. The authors should discuss the implications of their new findings. 
  3. IMvigor210 & GSE78220 data: I suggest to include these findings and discuss the different tumor entities as a limitation. However, I would disclose the different tumor entities in the methods and results sections. 
  4. In the current form, the paper is too extensive. I suggest that the authors substantially shorten the manuscript and focus on their main findings. Some analyses an figures could be transferred to supplementary material.
  5. There is still some overinterpretation of the results (including but not limited to "Patients with a lower PyroScore are more sensitive to targeted therapy and immunotherapy." in the abstract). The authors should still emphasize more clearly that this is in silico data that warrants further validation in vitro, in vivo and in prospective trials. 

Minor comments:

  1. Language still needs substantial revision even if English editing has been done by MDPI. For instance, there are typos including "patines" in line 40 and also style could be further improved.
  2. Figures are still almost illegible.
  3. Introduction: The fact that "A phase III study of LGG patients treated with either TMZ alone or radiotherapy alone did not find a significant difference in progression-free survival time [4]." is not a basis for the statement that the value of TMZ is unclear in LGG. Indeed, TMZ is increasingly used over the PCV regimen due to its easier application and better side effect profile in LGG. 

Reviewer 2 Report

The authors addressed all concerns and questions; the manuscript presentation is significantly improved.

Author Response

Thanks.

Round 3

Reviewer 1 Report

The authors considerably improved their manuscript. Congratulations!

I have one last suggestion for Figure 11: I would group therapy response between CR/PR/SD on one side and PD on the other side, as this is clinically more relevant.